# Therapeutic Potential of Olfactory Ensheathing Cells and Adipose-Derived Stem Cells in Osteoarthritis: Insights from Preclinical Studies

**DOI:** 10.3390/cells13151250

**Published:** 2024-07-25

**Authors:** Yu-Hsun Chang, Kun-Chi Wu, Chih-Jung Hsu, Tsui-Chin Tu, Mei-Chun Liu, Raymond Yuh-Shyan Chiang, Dah-Ching Ding

**Affiliations:** 1Department of Pediatrics, Hualien Tzu Chi Hospital, Buddhist Tzu Chi Medical Foundation, Tzu Chi University, Hualien 970, Taiwan; cyh0515@gmail.com; 2Department of Orthopedics, Hualien Tzu Chi Hospital, Buddhist Tzu Chi Medical Foundation, Tzu Chi University, Hualien 970, Taiwan; drwukunchi@yahoo.com.tw; 3Top Medical Biomedical Co., Ltd., Yilan City 260, Taiwan; ian@bio-nin.com (C.-J.H.); tree@bio-nin.com (T.-C.T.); may@bio-nin.com (M.-C.L.); 4Department of Obstetrics and Gynecology, Hualien Tzu Chi Hospital, Buddhist Tzu Chi Medical Foundation, Tzu Chi University, Hualien 970, Taiwan; raymond880106@gmail.com; 5Institute of Medical Sciences, Tzu Chi University, Hualien 970, Taiwan

**Keywords:** adipose stem cells, MMP13, olfactory ensheathing cells, osteoarthritis, type II collagen

## Abstract

Olfactory-ensheathing cells (OECs) are known for their role in neuronal regeneration and potential to promote tissue repair. Adipose-derived stem cells (ADSCs), characterized by mesenchymal stem cell (MSC) traits, display a fibroblast-like morphology and express MSC surface markers, making them suitable for regenerative therapies for osteoarthritis (OA). In this study, OECs and ADSCs were derived from tissues and characterized for their morphology, surface marker expression, and differentiation capabilities. Collagenase-induced OA was created in 10-week-old C57BL/6 mice, followed by intra-articular injections of ADSCs (1 × 10^5^), OECs (1 × 10^5^), or a higher dose of OECs (5 × 10^5^). Therapeutic efficacy was evaluated using rotarod performance tests, MRI, histology, and immunohistochemistry. Both cell types exhibited typical MSC characteristics and successfully differentiated into adipocytes, osteoblasts, and chondrocytes, confirmed by gene expression and staining. Transplantation significantly improved rotarod performance and preserved cartilage integrity, as seen in MRI and histology, with reduced cartilage destruction and increased chondrocytes. Immunohistochemistry showed elevated type II collagen and aggrecan in treated joints, indicating hyaline cartilage formation, and reduced MMP13 and IL-1β expression, suggesting decreased inflammation and catabolic activity. These findings highlight the regenerative potential of OECs and ADSCs in treating OA by preserving cartilage, promoting chondrocyte proliferation, and reducing inflammation. Further research is needed to optimize delivery methods and evaluate long-term clinical outcomes.

## 1. Introduction

Osteoarthritis (OA) is a degenerative joint disease characterized by cartilage breakdown, leading to pain, stiffness, swelling, and decreased mobility [1,2]. It is the most common form of arthritis, affecting millions of people worldwide [3]. The incidence of OA increases with age and is influenced by several factors, including genetics, obesity, joint injuries, repetitive stress on the joints, and certain metabolic diseases [4,5]. The latest population data indicate that OA affects over 300 million people globally, with a significant burden in older adults [6]. In the United States alone, it is estimated that over 32 million adults have OA, with a higher prevalence among women compared to men [7].

Factors that increase the incidence of OA include age, as the risk of OA increases significantly with age, particularly affecting individuals over 50 years old [8]; genetics, as a family history of OA can predispose individuals to develop the condition [9]; obesity, as excess body weight increases stress on weight-bearing joints, such as the knees and hips, accelerating the wear and tear of cartilage [10,11]; joint injuries, as previous joint injuries, including fractures, dislocations, and ligament tears, can lead to the development of OA [12]; repetitive stress, as occupations or activities that involve repetitive motion or stress on specific joints can contribute to OA development [13]; gender, as women are more likely to develop OA, particularly after menopause, possibly due to hormonal changes [14]; and metabolic diseases, as conditions such as diabetes and hemochromatosis can influence the onset of OA [15,16].

Diagnosing OA in its early stages presents several challenges [17,18]. Early OA may not show significant changes on conventional radiographic imaging, making detecting it difficult [19]. Symptoms like joint pain and stiffness are often mild and attributed to aging or other non-specific causes [20]. Moreover, a lack of specific biomarkers for early OA complicates early diagnosis [21]. Advances in imaging technologies, such as magnetic resonance imaging (MRI) and ultrasound, offer more detailed visualization of joint structures and can detect early cartilage changes [22,23,24]. Still, they are not routinely used in clinical practice, due to cost and accessibility.

Furthermore, there is a need for improved understanding and identification of molecular and biochemical markers that could facilitate early diagnosis and monitoring of OA progression [25,26]. Early intervention is crucial to managing OA effectively and improving patient outcomes [27,28,29]. Research into novel diagnostic tools and regenerative therapies, such as stem cell treatments, holds promise for advancing the management of OA [27,30,31].

Despite advancements in surgical techniques, conventional methods such as osteochondral transplantation may not be suitable for addressing large cartilage defects typical of OA, especially when the damage extends to multiple areas of the joint surface [32]. Moreover, these procedures may not prevent the progression of OA or address the associated inflammatory responses within the joint microenvironment [33]. Stem cell therapy, particularly mesenchymal stem cells (MSCs), holds promise for regenerating damaged cartilage and mitigating inflammation [33]. MSCs, derived from various sources such as bone marrow, adipose tissue, and umbilical cord blood, can differentiate into chondrocytes and contribute to tissue repair [31,34]. Additionally, they exhibit immunomodulatory properties, interact with immune cells, and release anti-inflammatory cytokines and growth factors that promote tissue regeneration and inhibit the inflammatory cascade [33,34].

In addition to their regenerative potential, MSCs have been explored for their ability to modulate immune responses in OA joints, potentially halting disease progression and providing long-term relief [33,35]. Olfactory ensheathing cells (OECs), primarily recognized for their role in repairing the nervous system, present a novel avenue for OA therapy owing to their unique immunomodulatory properties [36]. OECs, such as macrophages and T cells, interact with resident immune cells, regulating their activity and dampening inflammation [36]. However, the efficacy of OECs in OA treatment requires further investigation, particularly regarding their potential to promote cartilage repair and modulate immune responses in the joint microenvironment.

The development of effective therapies relies on suitable animal models that mimic pathogenesis [37,38]. Recent models, including injecting metabolic inhibitors or inducing ligament damage in animals, closely resemble human OA, aiding the development of new treatments, including stem cell therapy [37,38].

The potential of OECs and adipose-derived stem cells (ADSCs) for treating OA is being explored in this context. These cells exhibit regenerative properties that may help preserve cartilage integrity, promote chondrocyte proliferation, and reduce inflammation and catabolic activity within the joints, offering a novel approach to managing OA and improving the quality of life for affected individuals.

This study aimed to evaluate the ability of OECs and ADSCs to recover from cartilage destruction in a mouse model of OA.

## 2. Materials and Methods

### 2.1. ADSCs and OECs, Culture and Identification of Their Characteristics

The detailed protocol for deriving MSCs was aligned with our previously published paper [31]. We used human OECs from a biotechnology company (Top Medical Biomedical Co., Ltd., Yilan, Taiwan). The Research Ethics Committee of Hualien Tzu Chi Hospital approved the experimental protocol (approval number IRB112-126-C).

The olfactory nasal mucosal tissues were collected from the root of the medial aspect of the middle turbinate, washed three times at room temperature with wash solution (Gentamicin and Amphotericin B in DPBS), cut into 0.5 mm^3^ to 1 mm^3^ pieces, and then washed three times at room temperature with wash solution. The pieces were then placed in a T25 flask. Dulbecco’s modified Eagle’s medium F12, DMEM/F12 (Gibco, Thermo Fisher Scientific Inc., Waltham, MA, USA) containing 5% nLiven PR (Sexton Biotechnologies Inc., Indianapolis, IN, USA), 1% GlutaMAX (Gibco, Thermo Fisher Scientific Inc.), 10 ng/mL FGF2 (R&D Systems, Minneapolis, MN, USA), 20 µg/mL Gentamicin, and 0.125 µg/mL Amphotericin B was added. The tissues were incubated at 37 °C in 5% CO_2_. After seven days, the culture medium was changed to one without antibiotics. Seven to ten days later, the cells began to invade the flask, and after two to three weeks, the cells were confluent. When confluence had reached 90%, the cells were passaged and transferred to new flasks. Cells were plated at 5000 cells/flask in DMEM/F12 complete medium.

The human ADSCs were cultured in DMEM-F12 (Caisson Laboratories, Inc., Smithfield, UT, USA) containing 10% fetal bovine serum (Biological Industries, Kibbutz Beit Haemek, Israel), 0.2 mmol/L L-ascorbic acid 2-phosphate sesquimagnesium salt hydrate, 2 mmol/L N-acetyl-L-cysteine (Sigma-Aldrich, St. Louis, MO, USA), and 1% penicillin/streptomycin (Sigma-Aldrich) at 37 °C in a 95% air/5% CO_2_ humidified atmosphere.

ADSCs and OECs were characterized as described in our previous report, using surface markers detected by flow cytometry. According to a previous report, ADSC and OEC differentiation assays were performed to confirm the typical MSC characteristics [31].

### 2.2. Cell Proliferation Analysis

OECs from six donors across passages 1, 2, and 3 were used to construct growth curves. To count cell numbers using a hemocytometer, the device was prepared by cleaning it and placing a coverslip on the chamber. The cell sample was diluted and about 10 µL was loaded into the chamber. Under a microscope, the cells were counted in the four corner squares and the center square using a consistent method. The counts were averaged and multiplied by the dilution factor and chamber volume to determine the cell concentration in cells/mL, ensuring accurate results by cleaning the hemocytometer between uses. The counting formula was cell concentration = total count of cells × dilution factor/volume of counting chamber.

### 2.3. Flow Cytometry

Flow cytometry was used to assess the surface marker expression of ADSCs at passages 3–4 and OECs at passages 3–8. OECs and ADSCs were isolated by treatment with phosphate-buffered saline (PBS) containing Accutase (Millipore, Billerica, MA, USA). They were subsequently washed with PBS containing 2% bovine serum albumin and 0.1% sodium azide (Sigma-Aldrich). The cells were then incubated with primary antibodies labeled with either phycoerythrin or fluorescein isothiocyanate targeting surface markers, such as CD14, CD34, CD44, CD45, CD73, CD90, CD105, HLA-ABC, and HLA-DR (BD Biosciences, Franklin Lakes, NJ, USA). The analysis used a Becton Dickinson flow cytometer (Becton Dickinson, San Jose, CA, USA).

### 2.4. Trilineage Differentiation

#### 2.4.1. Adipogenesis

The adipogenic medium consisted of DMEM with 10% FBS, 0.5 mmol/L isobutylmethylxanthine, 5 µg/mL insulin, 60 μmol/L indomethacin, and 1 µmol/L dexamethasone (Sigma-Aldrich). ADSCs were plated at a density of 5 × 10^4^ cells/well in a 12-well plate and cultured in adipogenic medium for 14 days, with the medium replaced every 3 days. Differentiated adipocytes were stained using Oil Red O (Sigma-Aldrich), and images of the stained cells were captured using a microscope (Nikon, Tokyo, Japan). For OECs (5 × 10^4^) adipogenesis, upon reaching 80% cell confluency, the medium was replaced with pre-warmed Complete Adipogenesis Differentiation Medium (StemPro^®^ Gibco, Thermo Fisher Scientific Inc.). Cultures were re-fed every to 3–4 days. After 14 d of culture, the cells were stained with Oil Red O (Sigma-Aldrich) to detect adipogenesis. The cells were fixed in 4% PFA (Santa Cruz Biotechnology, Dallas, TX, USA) for 30 min at room temperature, washed with PBS, stained with Oil Red O (Sigma) working solution for 15 min at room temperature, washed thrice with deionized water, and incubated in PBS. The stained matrices were observed under a microscope at various magnifications.

#### 2.4.2. Osteogenesis

The osteogenic medium was formulated with DMEM supplemented with 10% FBS, 10 mmol/L β-glycerol phosphates, 0.1 µmol/L dexamethasone, and 50 μmol/L ascorbic acid (Sigma). ADSCs were seeded at a density of 1 × 10^4^ cells/well in a 12-well plate and cultured in an osteogenic medium for 14 days, with medium replenishment every 3 days. Osteoblast-like cells were visualized using Alizarin Red staining (Sigma-Aldrich), and images were captured using a microscope (Nikon). For OEC (5 × 10^4^) osteogenesis, upon reaching 80% cell confluency, the medium was replaced with pre-warmed Complete Osteogenesis Differentiation Medium (StemPro^®^ Gibco, Thermo Fisher Scientific Inc.). Cultures were re-fed every to 3–4 days. After 14 d of culture, the cells were processed for ALP (TaKaRa Bio, Kusatsu, Japan) and hematoxylin (Sigma-Aldrich) staining to detect osteogenesis. The staining procedure used to detect osteogenesis was similar to that used to detect adipogenesis.

#### 2.4.3. Chondrogenesis

The chondrogenic medium consisted of DMEM supplemented with 10% FBS, 6.25 μg/mL insulin, 10 ng/mL transforming growth factor-β1, and 50 μg/mL ascorbic acid-2-phosphate (Sigma). Chondrogenesis was induced using the pellet culture method, where 1 × 10^6^ ADSCs were seeded in a 15 mL conical tube (BD Biosciences) containing 2 mL of chondrogenic medium for 21 days, with medium replacement every 2 days. After culture, the resulting pellet was photographed and fixed in 4% paraformaldehyde at 4 °C for 24 h. The pellets were washed with PBS and transferred to a 70% ethanol solution. Histological analyses were used to characterize the differentiated chondrocytes, including hematoxylin and eosin (H & E) and safranin O staining. The pellets were embedded in paraffin and cut into sections. Immunohistochemical analyses were performed for type II collagen and aggrecan.

For OECs (1 × 10^6^) chondrogenesis, OECs were seeded in a 15 mL tube and cultured at 37 °C in a humidified atmosphere with 5% CO_2_ in DMEM/F12 complete medium. After 2–3 days, the medium was replaced with pre-warmed Complete Chondrogenesis Differentiation Medium (StemPro^®^ Gibco, Thermo Fisher Scientific Inc.). Cultures were re-fed every 3–4 days. Chondrogenic pellets were harvested after 21 days of culture. The pellets were fixed in 4% PFA overnight at 4 °C. The PFA was then replaced with 70% ethanol, and the pellets were paraffin embedded for safranin O staining.

### 2.5. Karyotyping

After collecting 2 × 10^6^ OECs at passage eight, karyotyping was performed at the Gene and Stem Cell Manufacturing Center of Hualien Tzu Chi Hospital. Chromosome analysis was performed using G-bands according to the guidelines of the International System for Chromosome Nomenclature 2009 (ISCN 2009).

### 2.6. Quantitative RT-PCR

Following the trilineage differentiation of ADSCs, total RNA was extracted from the cells using an RNeasy Protect Mini Kit with an on-column RNase-free DNase treatment (Qiagen, Hilden, Germany). RNA elution was performed using 30 mL of RNase-free water. Subsequently, 8 mL of the eluate was reverse transcribed using a SuperScript III One-Step RT-PCR Kit (Invitrogen, Grand Island, NY, USA) to synthesize cDNA. Real-time PCR amplification was conducted using FastStart SYBR Green QPCR Master Mix (Roche, Indianapolis, IN, USA) on a quantitative PCR detection system (ABI Step One Plus system, Applied Biosystems, Foster City, CA, USA), with 2 mL of the cDNA product. The primer sequences used for the PCR amplification are listed in Table 1 (all at a final concentration of 150 nM).

### 2.7. Collagenase-Induced Osteoarthritis Model

The Animal Care Committee of Hualien Tzu Chi Hospital approved the experimental protocol. Relevant guidelines and regulations for performing animal studies were followed.

A collagenase-induced OA model was established, as previously described [39]. Ten-week-old C57BL/6 mice that had undergone rotarod training were used in this study. The knee joints of these mice were injected once intra-articularly through the patellar ligament with 12 U of collagenase VII (*Clostridium histolyticum*; Sigma–Aldrich) in 10 μL saline on day 0. The mice were treated with 10 μL saline, ADSCs, or OECs on day 7.

These mice were divided into the following groups: Normal group (three mice) with 10 μL saline treatment. Negative control group (three mice): collagenase-induced OA mice with 10 μL saline treatment. The positive control group (six mice) comprised collagenase-induced OA mice treated with 1 × 10^5^ ADSCs. Study group 1 (6 mice): collagenase-induced OA mice treated with 1 × 10^5^ OECs. Study group 2 (6 mice): collagenase-induced OA mice treated with 5 × 10^5^ OECs.

### 2.8. Rotarod Test

Behavioral assessments were conducted during the light phase at 7-day intervals post-OA induction. Mice underwent forced ambulation via the Rotarod test (3376-4R, TSE Systems, Chesterfield, MO, USA) after a 30 min habituation period. Before testing, the mice underwent three days of rotarod training, and only those that passed were included. Functional assessments were performed on day 0 before saline/collagenase VII treatment, day 7 before cell transplantation, and days 14, 21, and 28 after transplantation. Mice were subjected to a rotarod performance test. The results were compared with the baseline durations, with data presented as the mean time on the rotating bar over five trials, with a maximum duration of 1200 s per trial at a set speed of 20 rpm.

### 2.9. Magnetic Resonance Imaging

Before sacrifice, mice in the control and OEC-treated groups underwent magnetic resonance imaging (MRI; Bruker BioSpec 70/20, Billerica, MA, USA) of the right knees. The mice were then euthanized. The severity of each joint was evaluated and classified into normal, mild, moderate, and severe by an experienced orthopedic physician (Wu KC).

### 2.10. Tissue Harvest

#### 2.10.1. Macroscopic Examination

After the mice were euthanized, their joint surfaces were grossly examined. The distal femur and proximal tibial surfaces were exposed and examined macroscopically.

#### 2.10.2. Histological Evaluation

The distal femur and the proximal tibial plateau were removed. After fixation with 10% buffered formalin (Sigma) for 48 h, the specimens were decalcified with 10% EDTA (Gibco, Grand Island, NY, USA) for two weeks and cut into four pieces. All pieces were embedded in paraffin. Serial sagittal sections were prepared and stained with hematoxylin and eosin (H & E, Sigma) and Safranin-O (Sigma). Histological changes were directly observed under a microscope. Quantitative evaluation of cartilage repair was performed using cartilage thickness and using the International Cartilage Repair Society (ICRS) scoring system [40]. We randomly selected three slides from each of the four mice, with and without OECs or ADSCs transplantation and normal control. Using the ICRS score, we evaluated the cartilage from the mice across six categories: surface, matrix, cell distribution, cell population viability, subchondral bone, and cartilage mineralization. The scores ranged from 0 (worse damaged cartilage) to 18 (normal hyaline cartilage).

### 2.11. Immunohistochemistry

The articular sections of the tibia were rehydrated and blocked with 3% hydrogen peroxide (Sigma-Aldrich). Paraffin sections were cut into serial sections with a thickness of 5 μm. Subsequently, the sections were blocked with Ultra V block (Thermo Scientific) for 10 min and then incubated with primary antibodies against type II collagen and aggrecan (1:200, GeneTex, Irvine, CA, USA) at 37 °C for 4 h. The secondary antibodies used were biotin-labeled goat anti-rabbit immunoglobulin (Dako, Carpinteria, CA, USA) and horseradish peroxidase-conjugated streptavidin (Biocare Medical), which were incubated for 30 min. Finally, the sections were stained with 3,3-diaminobenzidine solution, and hematoxylin (Sigma) was used to counterstain the slides. ImageJ software (National Institutes of Health, Bethesda, MD, USA) was used to quantify the immunohistochemical intensity of the cartilage.

Immunohistochemistry staining of IL1-β (1:200, Abbexa, Cambridge, UK) and MMP13 (1:200, Novus Biologicals, Centennial, CO, USA) was performed to assess the inflammatory and catabolic status of the joints in each group. Fifty cells were randomly counted from three areas, and the average number of positively stained cells was determined.

### 2.12. Statistical Analysis

All data were expressed as median and range or mean ± standard deviation. Statistical comparisons of the histopathological grades among the four groups used non-parametric tests, such as the Mann–Whitney U test or ANOVA with post hoc analysis. Differences were considered significant when the *p*-value was <0.05. All the statistical analyses were performed using SPSS version 25 (IBM Corp., Armonk, NY, USA).

## 3. Results

### 3.1. ADSCs Present Typical MSC Characteristics

Morphology and MSC surface markers were used to identify MSC characteristics [41]. The ADSCs showed a fibroblast-like morphology (Figure 1A,B). The surface markers were positive for CD44, CD73, CD90, CD105, and HLA-ABC and negative for CD34, CD45, and HLA-DR (Figure 1C). According to these findings, the ADSCs were based on the characteristic MSC morphology and surface marker panels.

Differentiation of ADSCs into adipocytes, osteoblasts, and chondrocytes. We performed qRT-PCR to identify the adipogenic, osteogenic, and chondrogenic genes. Compared to undifferentiated ADSCs, the ADSC-differentiated adipocytes showed increased expression of *FABP4* and *PPARγ* (Figure 1D). ADSC-differentiated osteoblasts showed an increased expression of *APAL* and *RUNX2* (Figure 1E). ADSC-differentiated chondrocytes showed an increased expression of aggrecan and *COL2A1* (Figure 1F).

After 14 days of adipogenic differentiation, the ADSC-differentiated adipocytes showed positive Oil Red O staining and intracytoplasmic oil droplets (Figure 1G–H). In contrast, after 14 days of osteogenic differentiation, ADSC-differentiated osteoblasts showed positive Alizarin Red staining and intracellular mineral deposits (Figure 1I,J). Furthermore, after 21 days of chondrogenic differentiation, the ADSCs formed a pellet (Figure 1K) and were stained with hematoxylin and eosin (Figure 1L), which revealed differentiated chondrocytes. The pellets were also positive for safranin O (staining sulfated glycosaminoglycans) (Figure 1M). Immunohistochemistry staining was positive for aggrecan and type II collagen (Figure 1N,O).

### 3.2. OECs Present Typical MSC Characteristics

Olfactory nasal mucosal tissue was collected from the bottom of a conical tube (Figure 2A). After culturing, the attached cells grew from the tissues (Figure 2A, P0 Day 3), proliferated to form a large area of cells after day 6, and showed a fibroblast-like appearance (Figure 2A). The proliferation curves of OECs are shown in Figure 2B. Different OEC cell lines exhibited different proliferation rates from P1 to P3. The OEC surface markers were positive for CD73, CD90, and CD105 and negative for CD14, CD34, and CD45 (Figure 3A). Trilineage differentiation was achieved after adipogenesis (positive Oil Red O staining), osteogenesis (positive ALP staining), and chondrogenesis (positive safranin O staining) (Figure 3B). ADSCs were used as positive controls. A conventional karyotype analysis was performed in the eighth passage (Figure 3C). The OECs expanded in vitro and did not show chromosome elimination, displacement, or imbalances.

### 3.3. Rotarod Behavior after Treatment

The rotarod test was used to evaluate the walking capacity of mice following a knee injury and stem cell treatment. This test measured the ability of the mice to maintain an upright position on a rotating rod, and the duration of their stability was recorded.

The mice in each group were subjected to the rotarod performance test on days 0, 7, 14, 21, and 28. Compared with the OA control group, OA mice transplanted with 1 × 10^5^ ADSCs, 1 × 10^5^ OECs, or 5 × 10^5^ OECs showed significant improvement after day 14 (Figure 4).

### 3.4. Magnetic Resonance Image Study

Before sacrifice, all mice underwent an MRI of their right knees. The mice were then euthanized. The results for each group were evaluated (Figure 5). In the normal control group, the cartilage contour and joint space were well-maintained (Figure 5A). The cartilage surface was eroded in the OA group, and the joint space was narrowed (Figure 5B). After transplantation with 5 × 10^5^ OECs (Figure 5C), 1 × 10^5^ OECs (Figure 5D), or 1 × 10^5^ ADSCs (Figure 5E), the cartilage contour and joint space were well-maintained.

### 3.5. Histology of Joint Cartilage

The H&E staining showed less cartilage destruction in the joints that received 5 × 10^5^, 1 × 10^5^ OECs, and 1 × 10^5^ ADSCs therapy in the OA joints (Figure 6). Loss of cartilage and eroded cartilage surfaces were noted in the OA group (Figure 6B). A normal cartilage surface (Figure 6A) and good cartilage integrity were noted in the OECs- and ADSC-transplanted groups (Figure 6C–E).

The Safranin O staining showed more chondrocytes in the joints that received 5 × 10^5^, 1 × 10^5^ OECs, and 1 × 10^5^ ADSCs therapy in the OA joints (Figure 7). Safranin O staining intensity revealed proteoglycan content in the cartilage. Normal cartilage expression is illustrated in Figure 7A. The OEC- and ADSC-transplanted groups (Figure 7C–E) had a high intensity in cartilage compared to OA (Figure 7B).

The joints that received 5 × 10^5^, 1 × 10^5^ OECs, and 1 × 10^5^ ADSCs showed significantly higher International Cartilage Repair Society histological scores (mean = 16) than the OA joints (mean = 3) (*p* < 0.001) (Figure 8).

### 3.6. Increased Expression of Type II Collagen and Aggrecan after Transplantation of 5 × 10^5^, 1 × 10^5^ OECs, and 1 × 10^5^ ADSCs

In the immunohistochemical examination of the mouse knee specimens, type II collagen (Figure 9A,B) and aggrecan (Figure 9C,D) were evaluated. In the OA mice, the damaged cartilage appeared almost colorless, suggesting the absence of hyaline cartilage (Figure 9A,C). Conversely, knees transplanted with 5 × 10^5^, 1 × 10^5^ OECs, and 1 × 10^5^ ADSC-treated groups exhibited more widespread staining, indicating hyaline cartilage formation (Figure 9A,C). Analysis of type II collagen staining intensity (*n* = 4 per group, Figure 9B) revealed significantly higher levels in the 5 × 10^5^, 1 × 10^5^ OECs, and 1 × 10^5^ ADSC-treated groups than in the other two groups (*p* < 0.01). Additionally, aggrecan levels were significantly elevated in the 5 × 10^5^ (*p* < 0.01), 1 × 10^5^ OECs, and 1 × 10^5^ ADSC-treated groups (*n* = 4 per group) compared with those in the OA group (*p* < 0.05) (Figure 9D). A normal joint (saline control) was used as a positive control.

### 3.7. Decreased Expression of MMP13 and IL-1β in 5 × 10^5^, 1 × 10^5^ OEC, and 1 × 10^5^ ADSC-Treated Cartilages

To assess the reduction in catabolic effect and inflammation following OEC and ADSC treatment in cartilage, IHC was conducted targeting MMP13 and IL-1β (Figure 10). Following the transplantation of 5 × 10^5^, 1 × 10^5^ OECs, and 1 × 10^5^ ADSCs into OA mice, the MMP13 expression in the cartilage was significantly reduced (*p* < 0.01) (Figure 10A,B, *n* = 4). Similarly, IL-1β levels showed a significant decrease in cartilage post-transplantation of 5 × 10^5^, 1 × 10^5^ OECs, and 1 × 10^5^ ADSCs (*p* < 0.001) (Figure 10C,D, *n* = 4).

## 4. Discussion

OECs and ADSCs exhibited typical MSC characteristics, displaying a fibroblast-like morphology and expressing CD44, CD73, CD90, CD105, and HLA-ABC but lacking CD34, CD45, and HLA-DR. OECs and ADSCs successfully differentiated into adipocytes, osteoblasts, and chondrocytes, as confirmed by the increased expression of lineage-specific genes and positive staining for the respective markers. In a collagenase-induced osteoarthritis model, OEC and ADSC transplantation substantially improved rotarod performance and maintained cartilage integrity, as evidenced by MRI and histological analyses. OECs and ADSCs showed increased type II collagen and aggrecan expression, indicating their potential to increase chondrogenic activity in OA cartilage. Moreover, OECs and ADSCs reduced MMP13 and IL-1β expression, suggesting potential for suppressing inflammation and catabolic activity in OA cartilage.

Stem cell therapy holds substantial promise for OA treatment, with ADSCs and OECs emerging as promising therapeutic approaches. ADSCs can differentiate into various cell types involved in cartilage regeneration, including chondrocytes, responsible for producing the cartilage matrix [42]. ADSCs exert anti-inflammatory and immunomodulatory effects [43]. ADSCs can be induced to differentiate into chondrocyte-like cells in vitro and in vivo, promoting cartilage regeneration and repair in OA-affected joints [44]. ADSCs are derived from adipose tissue and can be easily obtained through minimally invasive procedures, such as liposuction, making ADSC-based therapies relatively accessible and less invasive for patients [45]. OECs can support axonal growth and tissue remodeling, which may benefit joint repair [46]. OECs exhibit immunomodulatory properties similar to ADSCs [47], which could help mitigate inflammation within OA joints and promote a more favorable environment for tissue healing. OECs have been shown to secrete various growth and neurotrophic factors that stimulate the regeneration of damaged cartilage tissue in OA joints [47]. OECs are less likely to induce immune rejection because of their low immunogenicity, making them suitable for allogeneic transplantation [48]. OECs can be harvested from the olfactory bulb or mucosa, which are relatively accessible compared with other types of neural stem cells, facilitating their isolation and transplantation for therapeutic purposes [49]. In summary, both OECs and ADSCs offer unique advantages for OA treatment, including regenerative potential, anti-inflammatory effects, and immunomodulatory properties. Further research and clinical trials are required to fully elucidate the therapeutic mechanisms and optimize their efficacy in OA management.

Type II collagen and aggrecan are essential components of the hyaline cartilage, providing structural integrity and elasticity to tissues [50]. In healthy cartilage, type II collagen forms the predominant collagenous framework, ensuring tensile strength and mechanical stress resilience [50]. Aggrecan is a proteoglycan that contributes to cartilage hydration and compression resistance [51]. Post-transplantation, the observed increase in type II collagen and aggrecan expression suggests stimulation of chondrogenic differentiation and extracellular matrix synthesis within the damaged cartilage. OECs and ADSCs likely promote chondrocyte proliferation and differentiation, producing a new cartilaginous matrix rich in type II collagen and aggrecan. Moreover, the upregulation of type II collagen and aggrecan signified a shift towards a more hyaline-like cartilage phenotype, which is desirable for cartilage repair in OA [52]. Hyaline cartilage, characterized by its smooth surface and homogeneous composition, possesses superior mechanical properties compared to fibrocartilage, often formed in OA [53]. This increase in type II collagen and aggrecan expression indicated the efficacy of stem cell therapy in promoting cartilage regeneration and restoring cartilage functionality and structural integrity. Ultimately, the enhanced deposition of type II collagen and aggrecan contributed to improving joint function and alleviating OA symptoms observed in the treated mice, highlighting the potential of stem cell transplantation as a therapeutic strategy for OA management.

MMP13 is a collagenase responsible for the degradation of type II collagen, the primary collagenous component of hyaline cartilage [54]. Elevated levels of MMP13 in OA cartilage contribute to the destruction of the cartilage matrix, leading to joint degeneration and loss of function [54]. IL-1β, a pro-inflammatory cytokine, induces the expression of MMPs and other catabolic factors in chondrocytes, exacerbating cartilage degradation and perpetuating the inflammatory cascade in OA [55]. Following stem cell treatment, the observed decrease in MMP13 and IL-1β expression suggests suppressing catabolic and inflammatory pathways within the osteoarthritic joint. OECs and ADSCs likely exert their therapeutic effects by modulating the local microenvironment and attenuating the production and activity of MMPs and proinflammatory cytokines. This modulation may occur through paracrine signaling as stem cells release various growth factors, cytokines, and anti-inflammatory molecules that influence neighboring cells and immune responses [56]. By reducing MMP13 expression, stem cell therapy helps preserve the structural integrity of the cartilage by inhibiting collagen degradation [57]. Similarly, suppressing IL-1β signaling mitigates inflammation within the joint, thereby alleviating pain and slowing disease progression in OA [58]. The combined effect of decreased MMP13 and IL-1β expression contributes to maintaining cartilage homeostasis and promoting tissue repair in osteoarthritic joints. Ultimately, the downregulation of MMP13 and IL-1β reflects stem cell therapy’s anti-catabolic and anti-inflammatory properties, highlighting its potential as a promising approach for OA treatment. This reduction in catabolic and inflammatory mediators protects existing cartilage from further damage. It creates a conducive environment for tissue regeneration and repair, ultimately improving joint function and quality of life in patients with OA.

The strength of this study is that it included a thorough approach for evaluating the therapeutic potential of OECs and ADSCs in a collagenase-induced OA model. It utilized multiple assays to comprehensively evaluate treatment outcomes, including rotarod performance tests, MRI, histological analyses, and immunohistochemistry. Second, by using a collagenase-induced mouse model of OA, this study mimicked the key pathological features of human OA, enhancing the relevance and translatability of the findings to clinical settings. Third, we compared the therapeutic effects of OECs and ADSCs, two distinct types of stem cells, in treating OA. This comparative analysis assessed the relative efficacy of different cell types, providing valuable information for optimizing stem cell-based therapies for OA management. Fourth, the study included molecular analyses, such as gene expression profiling and immunohistochemistry, to elucidate the mechanisms underlying the observed therapeutic effects of stem cell transplantation. This molecular characterization enhances our understanding of how OECs and ADSCs influence cartilage regeneration, inflammation, and catabolism in OA joints.

The current study had several limitations. First, although the collagenase-induced OA model replicates certain aspects of human OA, it may not fully recapitulate the complexity and heterogeneity of human disease. Second, the study’s follow-up period may have been too short to fully assess stem cell therapy’s long-term effects and durability for OA treatment. Third, the findings of this study may be specific to the experimental conditions and animal models used, which limits their generalizability to other OA models or patient populations. Fourth, although this study provides insights into the molecular mechanisms underlying the therapeutic effects of OECs and ADSCs in OA, further mechanistic studies are warranted to fully elucidate the cellular interactions, signaling pathways, and tissue remodeling processes involved in stem-cell-mediated cartilage repair and inflammation modulation.

## 5. Conclusions

In conclusion, OECs and ADSCs demonstrated typical MSC characteristics and hold promise for OA therapy. The ability of MSCs to differentiate into adipocytes, osteoblasts, and chondrocytes highlights their regenerative potential. In a collagenase-induced OA model, the transplantation of OECs and ADSCs substantially improved joint function and preserved cartilage integrity. Furthermore, OECs and ADSCs alleviated inflammation and catabolic activity in osteoarthritic cartilage, as evidenced by reduced MMP13 and IL-1β expression.

Looking ahead, these findings highlight the therapeutic efficacy of OECs and ADSCs in OA treatment and pave the way for potential regenerative medicine interventions in various joint disorders. Continued research and clinical trials will be crucial in further elucidating the mechanisms underlying their therapeutic effects and optimizing their application in clinical settings, ultimately offering hope for patients suffering from OA and other degenerative joint conditions.

## Figures and Tables

**Figure 1 cells-13-01250-f001:**
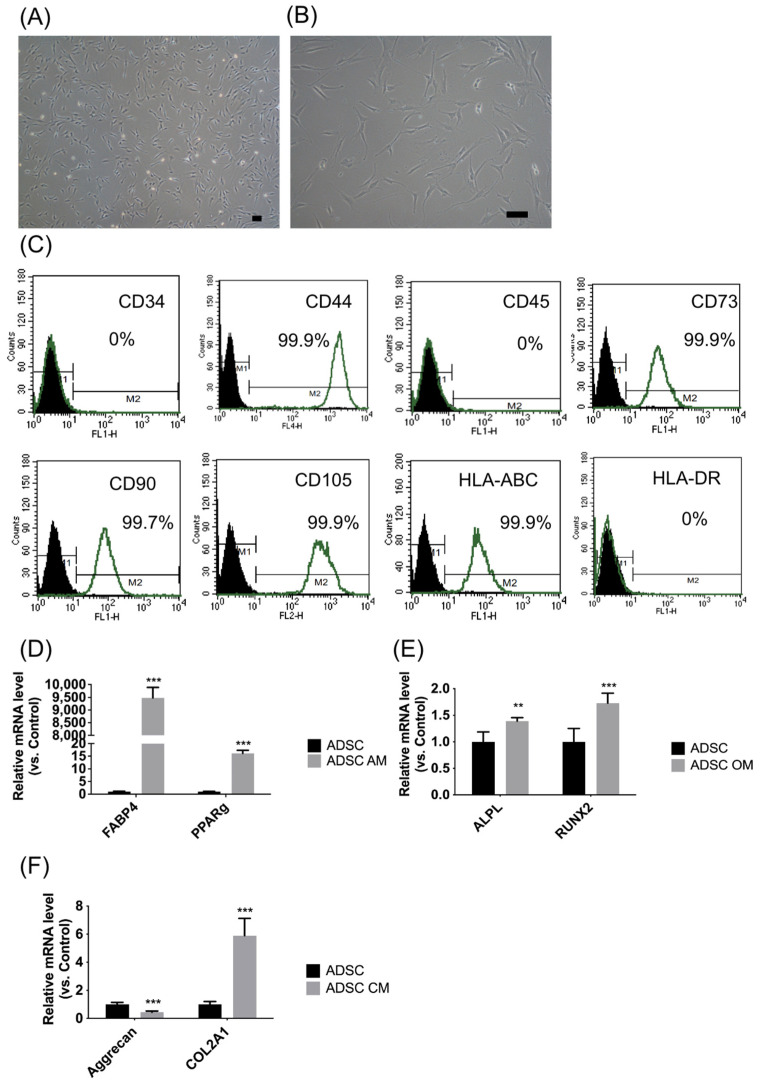
Adipose stem cell (ASC) characterization. (**A**,**B**) morphology of ADSC. Scale bar = 100 μm. (**C**) Flowcytometry of ASC. Positive for CD44, CD73, CD90, CD105, and HLA-ABC and negative for CD34, CD45, and HLA-DR. (**D**–**F**) qRT-PCR showed the gene expressions of adipogenesis (*FABP4*, *PPARγ*) (**D**), osteogenesis (*ALPL*, *RUNX2*) (**E**), and chondrogenesis (aggrecan and *COL2A1*) (**F**). (**G**) Oil red staining of differentiated cells after adipogenesis. (**H**) Quantification of Oil Red at OD 510. (**I**) Alizarin Red staining of differentiated cells after osteogenesis. (**J**) Quantification of Alizarin Red at OD552. (**K**–**O**) Characterization of differentiated cell after chondrogenesis. Pellet morphology (**K**), H & E (**L**), Safranin O (**M**), aggrecan (**N**), and type II collagen (**O**). ** *p* < 0.01, *** *p* < 0.001. Scale bar = 100 μm.

**Figure 2 cells-13-01250-f002:**
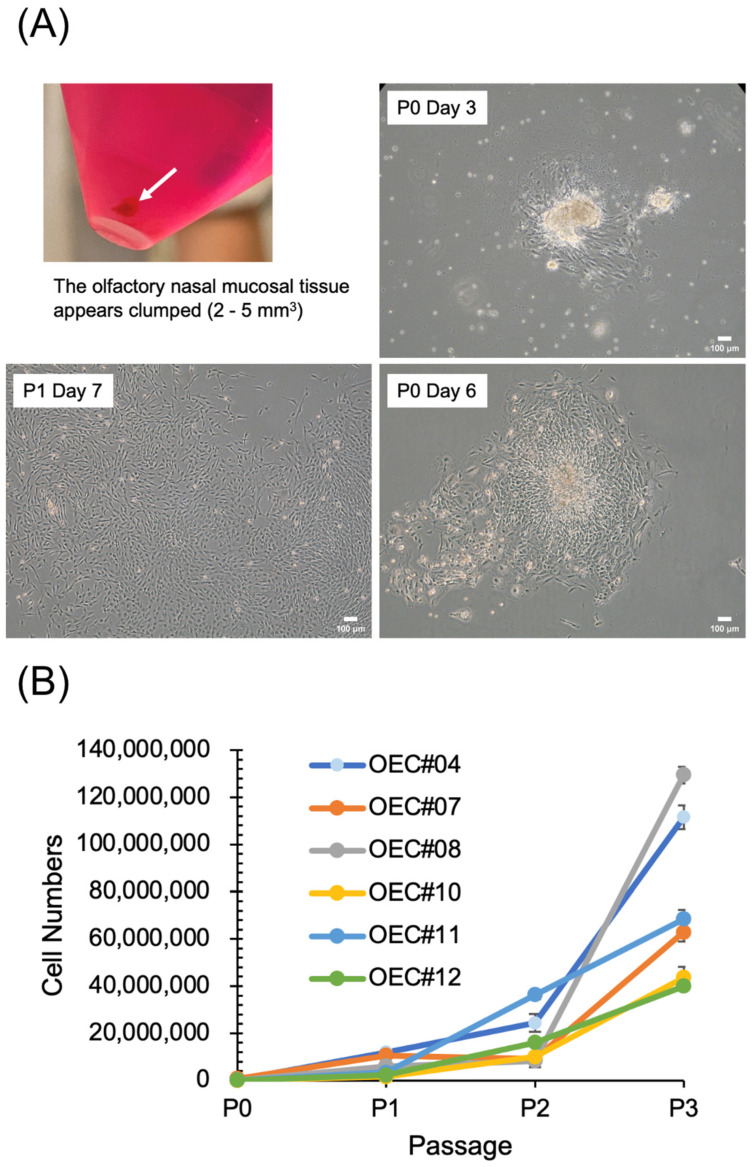
Characteristics of olfactory ensheathing cells (OECs). (**A**) The olfactory nasal mucosal tissue appears clumped (2–5 mm^3^) (arrow). Microscope morphological observation of OECs at passage (P)0 on days 3, 6, and P1 on day 7. Scale bar = 100 μm. (**B**) Proliferation curve analysis by cell counting (Hemocytometer) at P1 to P3 of OECs from the six donors.

**Figure 3 cells-13-01250-f003:**
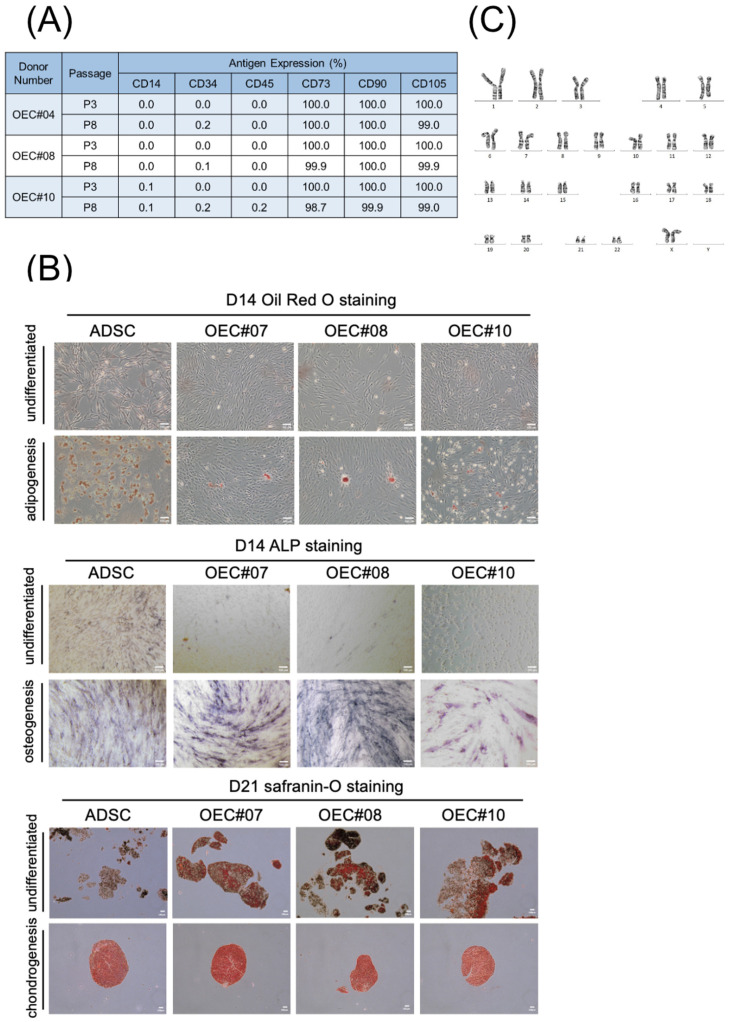
Characteristics of olfactory ensheathing cells (OECs). (**A**) The immunofluorescence analysis of OECs from three donors. Positive for CD73, CD90, CD105, and negative for CD14, CD34, and CD45. (**B**) Differentiating OECs from three donors and compared with adipose-derived stem cells (ADSCs). Oil Red O staining to detect adipogenesis differentiation. ALP staining to detect osteogenesis differentiation. Safranin O staining to detect chondrogenesis differentiation. Scale bar = 100 μm. (**C**) A conventional karyotype analysis was performed at the eighth passage. The number 1–22 are chromosomal number. X and Y are sex chromosomes. The OECs expanded in vitro and did not show chromosome elimination, displacement, or imbalances.

**Figure 4 cells-13-01250-f004:**
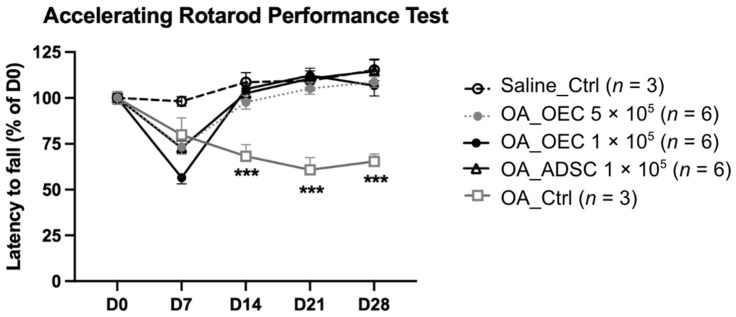
Rotarod performance test of OA mice treated with adipose tissue-derived stem cells (ADSCs) or olfactory ensheathing cells (OECs). Compared to the OA control group, we found that the OA mice transplanted with 1 × 10^5^ ADSCs, 1 × 10^5^ OECs, or 5 × 10^5^ OECs showed significant improvement after day 14. *** *p* < 0.001 compared to Saline_Ctrl, OA_OEC, and OA_ADSC.

**Figure 5 cells-13-01250-f005:**
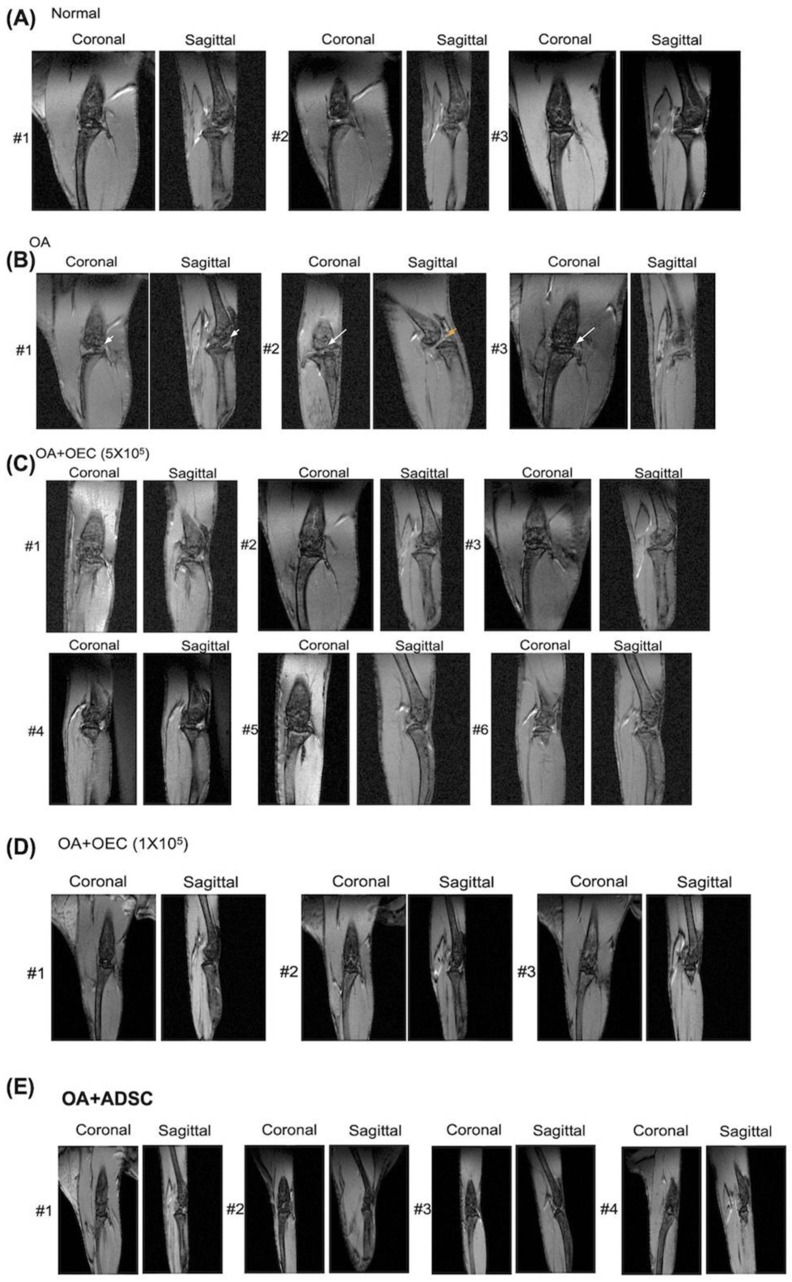
Magnetic resonance image study of cartilage after various treatments. (**A**) Normal-control mice (*n* = 3). The contour of cartilage and joint space was well-maintained. (**B**) Osteoarthritis (OA) mice with normal saline injection (*n* = 3). The cartilage surface was eroded, and the joint space was narrowed. Joint destruction (arrowhead). Joint space narrowing (yellow arrowhead). Joint transposition (arrow). (**C**) Cartilage in OA mice transplanted OECs (5 × 10^5^ cells) (*n* = 6). After OECs were transplanted, cartilage contour and joint space were well-maintained. (**D**) Cartilage in OA mice transplanted OECs (1 × 10^5^ cells) *(n* = 3). After OECs were transplanted, cartilage contour and joint space were well-maintained. (**E**) Cartilage in OA mice transplanted ASCs (1 × 10^5^ cells) (*n* = 4). After ASCs were transplanted, cartilage contour and joint space were well-maintained.

**Figure 6 cells-13-01250-f006:**
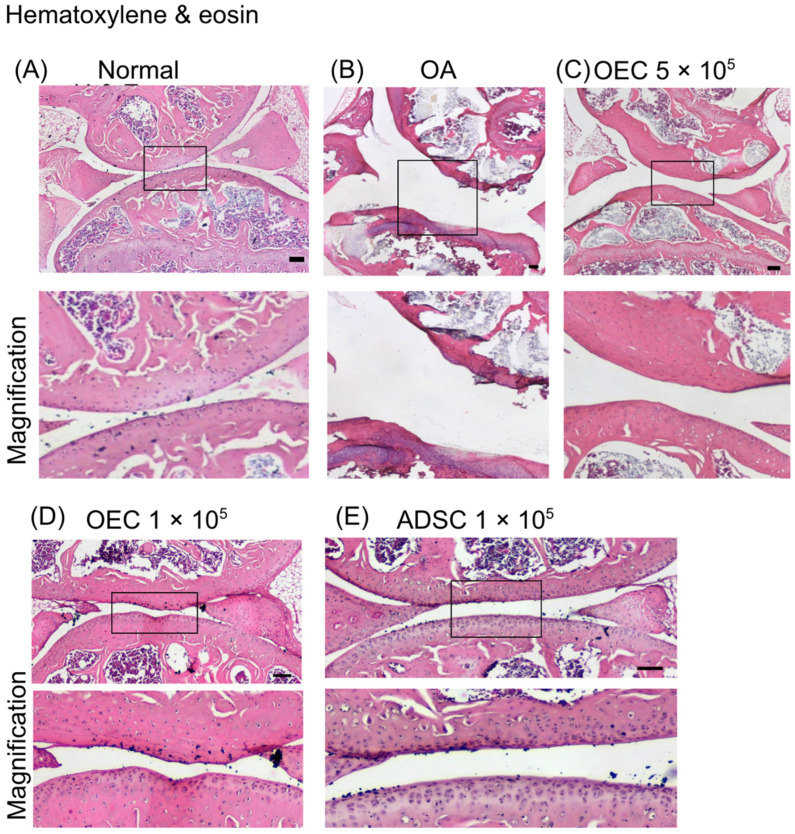
The histology of mouse osteoarthritis model after 28 days of experiments. A representative image is illustrated. (**A**) Normal knee joints (*n* = 4). (**B**) Osteoarthritis joints (*n* = 4). Cartilage was abrased to a thin layer. (**C**) Osteoarthritis mice received 5 × 10^5^ olfactory ensheathing cells (OECs) transplantation (*n* = 4). (**D**) Osteoarthritis mice received 1 × 10^5^ olfactory ensheathing cells (OECs) transplantation (*n* = 4). (**E**) Osteoarthritis mice received 1 × 10^5^ adipose-derived stem cells (ADSCs) transplantation (*n* = 4). The cartilage morphology in OECs and ADSCs transplant groups showed a normal appearance. The lower panel shows a magnified view of the upper panel.

**Figure 7 cells-13-01250-f007:**
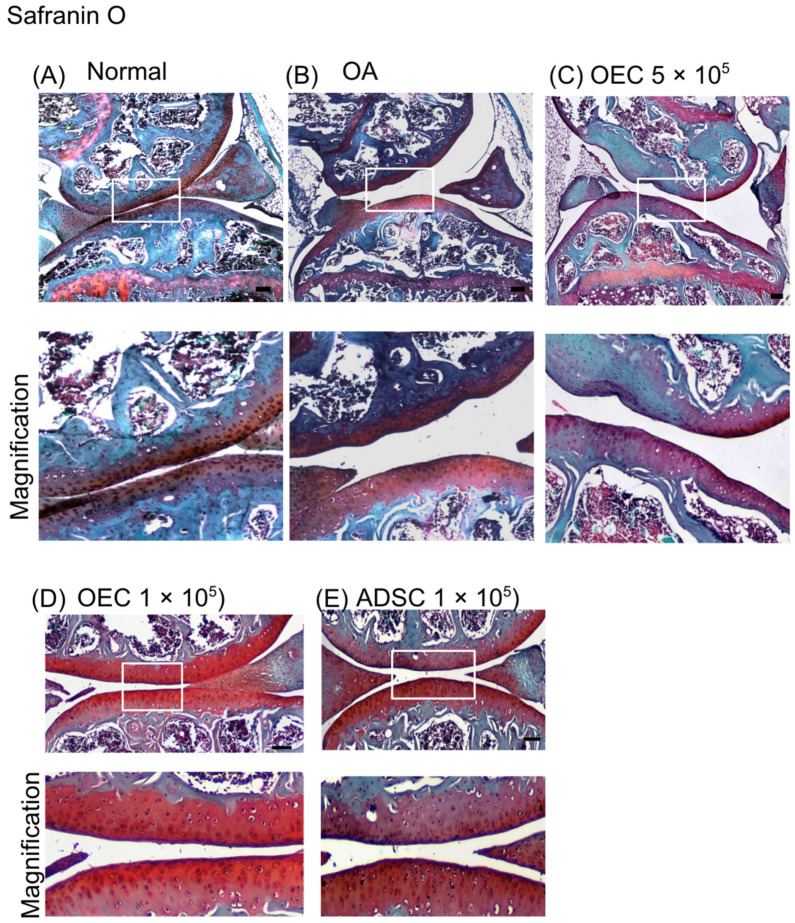
The histology of the mouse osteoarthritis model after 28 days of experiments (Safranin O). A representative image is illustrated. (**A**) Normal knee joints (*n* = 4). (**B**) Osteoarthritis joints (*n* = 4). Cartilage was abrased to a thin layer. (**C**) Osteoarthritis mice received 5 × 10^5^ olfactory ensheathing cells (OECs) transplantation (*n* = 4). (**D**) Osteoarthritis mice received 1 × 10^5^ olfactory ensheathing cells (OECs) transplantation (*n* = 4). (**E**) Osteoarthritis mice received 1 × 10^5^ adipose-derived stem cells (ADSCs) transplantation (*n* = 4). The lower panel shows a magnified view of the upper panel.

**Figure 8 cells-13-01250-f008:**
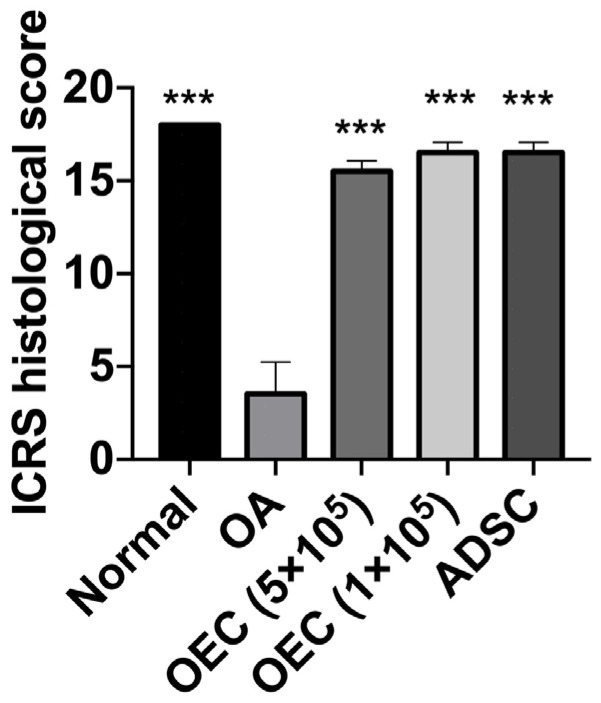
The International Cartilage Repair Society (ICRS) score of the mouse osteoarthritis model after 28 days of experiments. The ICRS score was improved after 5 × 10^5^ (*n* = 4), 1 × 10^5^ (*n* = 4) olfactory ensheathing cell (OEC), and 1 × 10^5^ adipose-derived stem cell (ADSCs, *n* = 4) transplantation. Normal group (*n* = 4). *** *p* < 0.001 compared to the osteoarthritis (OA) group.

**Figure 9 cells-13-01250-f009:**
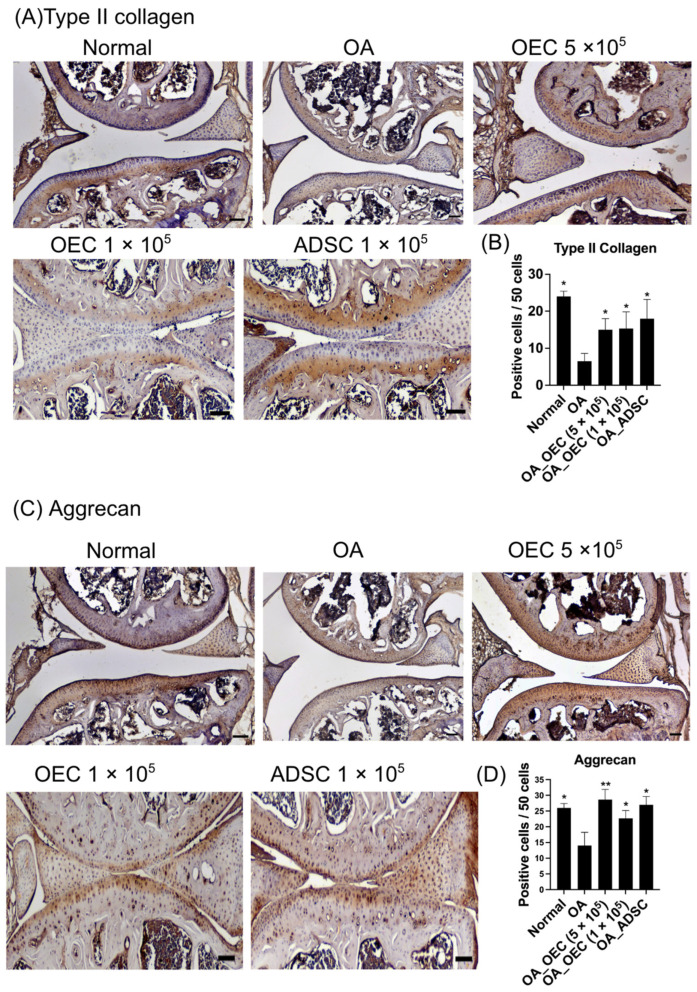
Immunohistochemistry of type II collagen and aggrecan of the mouse osteoarthritis model after 28 days of experiments. (**A**) A representative image of type II collagen is illustrated from the normal knee (*n* = 4), osteoarthritis (OA) (*n* = 4), 5 × 10^5^ olfactory ensheathing cells (OECs), 1 × 10^5^ OECs, and 1 × 10^5^ adipose-derived stem cells (ADSCs) (*n* = 4 in each group). (**B**) Quantitative analysis of positive staining cells in 50 cells in five fields. (**C**) A representative image of aggrecan is illustrated from the normal knee (*n* = 4), OA (*n* = 4), 5 × 10^5^ OECs, 1 × 10^5^ OECs, and 1 × 10^5^ ADSCs (*n* = 4 in each group). (**D**) Quantitative analysis of positive staining cells in 50 cells in five fields. * *p* < 0.05, ** *p* < 0.01.

**Figure 10 cells-13-01250-f010:**
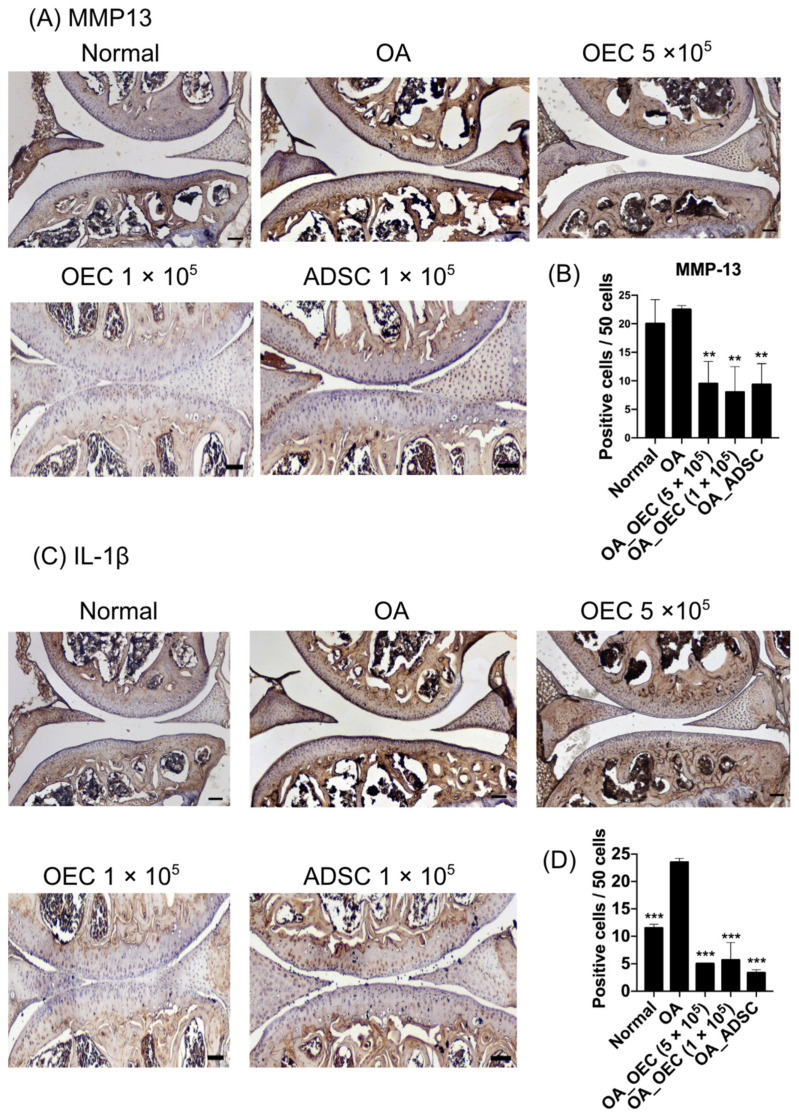
Immunohistochemistry of MMP13 and IL-1β of the mouse osteoarthritis model after 28 days of experiments. (**A**) A representative image of MMP13 is illustrated from the normal knee (*n* = 4), osteoarthritis (OA) (*n* = 4), 5 × 10^5^ olfactory ensheathing cells (OECs), 1 × 10^5^ OECs, and 1 × 10^5^ adipose-derived stem cells (ADSCs) (*n* = 5). (**B**) Quantitative analysis of positive staining cells in 50 cells in five fields. (**C**) A representative image of IL-1β is illustrated from the normal knee (*n* = 4), OA) (*n* = 4), 5 × 10^5^ OECs, 1 × 10^5^ OECs, and 1 × 10^5^ ADSCs (*n* = 4). (**D**) Quantitative analysis of positive staining cells in 50 cells in five fields. ** *p* < 0.01, *** *p* < 0.001.

**Table 1 cells-13-01250-t001:** Primer sequence.

Gene	Forward (5′->3′)	Reverse (5′->3′)
*PPAR-γ*	AGCCTCATGAAGAGCCTTCCA	TCCGGAAGAAACCCTTGCA
*FABP4*	ATGGGATGGAAAATCAACCA	GTGGAAGTGACGCCTTTCAT
*ALPL*	CCACGTCTTCACATTTGGTG	GCAGTGAAGGGCTTCTTGTC
*RUNX2*	CGGAATGCCTCTGCTGTTAT	TTCCCGAGGTCCATCTACTG
*COL2A1*	GGACTTTTCTCCCCTCT CT	GACCCGAAGGTCTTACAGGA
*ACAN*	GAGATGGAGGGTGAGGTC	GAGATGGAGGGTGAGGTC
*GAPDH*	GAAGGTGAAGGTCGGAGTC	GAAGATGGT GATGGGATTTC

Adipogenesis: *PPAR-γ*, *FABP4*, Osteogenesis: *ALPL*, *RUNX2*, Chondrogenesis: *COL2A1*, *ACAN*.

## Data Availability

All data are in the manuscript.

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
