# Peer review of "Therapeutic Potential of Olfactory Ensheathing Cells and Adipose-Derived Stem Cells in Osteoarthritis: Insights from Preclinical Studies"

_cells, 2024, doi:10.3390/cells13151250_

Round 1
Reviewer 1 Report
Comments and Suggestions for Authors
Overall assessment of the article:
The article Therapeutic Potential of Olfactory Ensheathing Cells and Adipose-Derived Stem Cells in Osteoarthritis: Insights from Preclinical Studies explores the regenerative potential of OECs and ADSCs in treating osteoarthritis (OA). Both cell types demonstrated typical MSC characteristics and successfully differentiated into adipocytes, osteoblasts, and chondrocytes. In a murine OA model, treated mice showed improved joint function, preserved cartilage integrity, and reduced inflammation. These findings highlight the promise of OECs and ADSCs in OA therapy, though further research is needed to optimize cell delivery methods and assess long-term outcomes in clinical settings.
The work is interesting and generally correctly written, it is a valuable source of up-to-date information but will require corrections and additions before proceeding further. I provide detailed notes and comments below.
Major comments:
The abstract is far too long. Please shorten the abstract retaining only the most important information and bring it in line with the requirements of the journal.
The introduction should be expanded to include more detailed information on osteoarthritis along with the latest literature. It presents only 12 papers which is far too small a number for a scientific article. I suggest adding to the introduction more detailed information about osteoarthritis, the factors that increase its incidence, the latest population data, information on the problems of diagnosing osteoarthritic changes in their early stages, along with the necessary recent literature.
The incidence of osteoarthritis is influenced by many factors, such as work, sports participation, musculoskeletal injuries, obesity and gender. Information on these topics, along with the necessary literature, should be added in the introduction, preferably in the first paragraphs. The authors can find useful information in the following papers: doi: 10.35784/acs-2023-40; DOI: 10.1056/NEJMcp1903768; DOI 10.3390/app10238312; DOI: 10.1056/NEJMcp051726; doi:10.35784/acs-2022-14; https://doi.org/10.1007/s10787-011-0118-0
Part A in Figure 1 is far too small making it unreadable. I suggest splitting the composite graphics into separate graphics (splitting will increase readability).
In conclusion, while the preclinical evidence is compelling, translating these findings into clinical practice will require extensive further research, including human trials to assess safety, efficacy, and long-term benefits. The insights gained from this study contribute valuable knowledge to the growing field of regenerative medicine and its application in osteoarthritis management.
After making appropriate corrections and additions to the content and literature, the work can be further processed and accepted for publication.
Author Response
Reviewer 1
Comments and Suggestions for Authors
Overall assessment of the article:
The article Therapeutic Potential of Olfactory Ensheathing Cells and Adipose-Derived Stem Cells in Osteoarthritis: Insights from Preclinical Studies explores the regenerative potential of OECs and ADSCs in treating osteoarthritis (OA). Both cell types demonstrated typical MSC characteristics and successfully differentiated into adipocytes, osteoblasts, and chondrocytes. In a murine OA model, treated mice showed improved joint function, preserved cartilage integrity, and reduced inflammation. These findings highlight the promise of OECs and ADSCs in OA therapy, though further research is needed to optimize cell delivery methods and assess long-term outcomes in clinical settings.
The work is interesting and generally correctly written, it is a valuable source of up-to-date information but will require corrections and additions before proceeding further. I provide detailed notes and comments below.
Major comments:
Comment 1: The abstract is far too long. Please shorten the abstract retaining only the most important information and bring it in line with the requirements of the journal.
Response 1: We thank the reviewer’s comment. We revised the abstract to shorten it to 200 words to fulfill the journal's requirement. (page 1, lines16-33, Abstract section)
Comment 2: The introduction should be expanded to include more detailed information on osteoarthritis along with the latest literature. It presents only 12 papers which is far too small a number for a scientific article. I suggest adding to the introduction more detailed information about osteoarthritis, the factors that increase its incidence, the latest population data, information on the problems of diagnosing osteoarthritic changes in their early stages, along with the necessary recent literature.
Response 2: We thank the reviewer’s comment. We have expanded the introduction to include the aspects that the reviewer suggested. We have added 31 updated references. (Page 1-2, lines 37-69, Introduction section)
Comment 3: The incidence of osteoarthritis is influenced by many factors, such as work, sports participation, musculoskeletal injuries, obesity and gender. Information on these topics, along with the necessary literature, should be added in the introduction, preferably in the first paragraphs. The authors can find useful information in the following papers: doi: 10.35784/acs-2023-40; DOI: 10.1056/NEJMcp1903768; DOI 10.3390/app10238312; DOI: 10.1056/NEJMcp051726; doi:10.35784/acs-2022-14; https://doi.org/10.1007/s10787-011-0118-0
Response 3: We thank the reviewer’s comment. We have added the references mentioned above to the introduction section.
Comment 4: Part A in Figure 1 is far too small making it unreadable. I suggest splitting the composite graphics into separate graphics (splitting will increase readability).
Response 4: We thank the reviewer’s comment. We have split parts A and B into Figure 1. The other part was changed to Figure 2 to make them look clear. (New Figure 2 and 3, Pages 10-11)
In conclusion, while the preclinical evidence is compelling, translating these findings into clinical practice will require extensive further research, including human trials to assess safety, efficacy, and long-term benefits. The insights gained from this study contribute valuable knowledge to the growing field of regenerative medicine and its application in osteoarthritis management.
After making appropriate corrections and additions to the content and literature, the work can be further processed and accepted for publication.
Reviewer 2 Report
Comments and Suggestions for Authors
I carefully read the manuscript “Therapeutic Potential of Olfactory Ensheathing Cells and Adipose-Derived Stem Cells in Osteoarthritis: Insights from Preclinical Studies” by Chang e al. The paper is easy to read and I have only minor suggestions that you should address before the publication.
Are ADSCs and OECs human or animal cells?
Line 103 Percentage of confluence should be indicated. 90%?
Line 104 It is not clear because you plated 3,000-7,000 cells/flask. Depends on what?
Lines 115-119 Cell proliferation assay should be better described
Line 153 For the sentence “Osteoblasts were visualized…” I suggest changing the term “osteoblasts” to “osteoblast-like” because these cells derive from differentiate ADSCs.
Line 159 Please check the sentence “The staining procedure used to detect adipogenesis was similar to that used to detect adipogenesis”
Line 170 Were pellets embedded in paraffin and cut into sections?
Line 213 Please check (yhree mice)
Line 307 The paragraph “ADSCs present typical MSC characteristics” should be reported before “OECs present typical MSC characteristics” because ADSCs serve as control and are typical mesenchymal stem cells. Moreover, this paragraph repots “an increased expression of aggrecan” (line 318) for ADSC-differentiated chondrocytes, but in figure 2F the graphic show a significant decrease in aggrecan RNA levels in ADSC CM compared to ADSCs! Check please. Also, Figure 2K is shown in the text, but is not explained.
The quality of figures is poor.
Figure 1 A Authors may include an arrow or a box to indicate the clumped olfactory nasal mucosal tissue
Figure 1B If I understand, you counted the cells after trypsinization and before the plating. So, how did you not get a nonlinear segment for the OEC#4 sample? Furthermore, error bars should be included in the curve, for example by counting the cells three time.
Figure 1D The microphotographs are too small and of low quality. I suggest you reduce the table (Figure 1C) and enlarge the photomicrographs. Moreover, the scale bar of Safranin-O is the same of the ALP and Oil-red O staining (100µm). Please check it.
Figure 2 Microphotographs in G, I, K, L, M, and N show differentiate ADSCs, undifferentiated cells should also be shown.
Figure 3 Do the asterisks refer to the comparison between groups and saline ctrl? The comparison should be reported in the figure legend as “ ***p<0.001 compared to the ctrl saline sample”. The mere indication “ctrl:control” is not indicative, also because you have a saline sample ctrl, an OA sample ctrl and a no sample ctrl alone.
Author Response
Reviewer 2
I carefully read the manuscript “Therapeutic Potential of Olfactory Ensheathing Cells and Adipose-Derived Stem Cells in Osteoarthritis: Insights from Preclinical Studies” by Chang e al. The paper is easy to read and I have only minor suggestions that you should address before the publication.
Comment 1: Are ADSCs and OECs human or animal cells?
Response 1: We thank the reviewer’s comment. Both cells are of human origin. We added “human” before introducing both cell lines in the Method section (page 3, lines 105 and 122).
Comment 2: Line 103 Percentage of confluence should be indicated. 90%?
Response 2: We thank the reviewer’s comment. We have added the percentage of confluence (90%). The statement reads as”When confluence reached 90%, the cells were passaged and transferred to new flasks.”(page 3, line 119)
Comment 3: Line 104 It is not clear because you plated 3,000-7,000 cells/flask. Depends on what?
Response 3: We thank the reviewer’s comment. We have changed it to 5,000 cells/flask. (page 3, line 120)
Comment 4: Lines 115-119 Cell proliferation assay should be better described
Response 4: We thank the reviewer’s comment. We have revised the whole paragraph. The statement read as”
Cell proliferation analysis
OECs from six donors across passages 1, 2, and 3 were used to construct growth curves. To count cell numbers using a hemocytometer, prepare the device by cleaning it and placing a coverslip on the chamber. Diluted the cell sample and loaded about 10 µL into the chamber. Under a microscope, count the cells in the four corner squares and the center square using a consistent method. Average the counts and multiply by the dilution factor and chamber volume to determine the cell concentration in cells/mL, ensuring accurate results by cleaning the hemocytometer between uses. The counting formula was cell concentration= total count of cells×dilution factor/ volume of counting chamber. “ (page 3, lines 133-140)
Comment 5: Line 153 For the sentence “Osteoblasts were visualized…” I suggest changing the term “osteoblasts” to “osteoblast-like” because these cells derive from differentiate ADSCs.
Response 5: We thank the reviewer’s comment. We have revised it accordingly. (page 4, line 175)
Comment 6: Line 159 Please check the sentence “The staining procedure used to detect adipogenesis was similar to that used to detect adipogenesis”
Response 6: We thank the reviewer’s comment. We have revised the first “adipogenesis” to “osteogenesis”. (page 4, line 181)
Comment 7: Line 170 Were pellets embedded in paraffin and cut into sections?
Response 7: We thank the reviewer’s comment. Yes, the pellets were embedded in paraffin and cut into sections. We have added the description to the methods. The statement reads as “The pellets were embedded in paraffin and cut into sections”. (page 4, lines 193-194)
Comment 8: Line 213 Please check (yhree mice)
Response 8: We thank the reviewer’s comment. We have revised “yhree” to “three”. (page 5, line 236)
Comment 9: Line 307 The paragraph “ADSCs present typical MSC characteristics” should be reported before “OECs present typical MSC characteristics” because ADSCs serve as control and are typical mesenchymal stem cells. Moreover, this paragraph repots “an increased expression of aggrecan” (line 318) for ADSC-differentiated chondrocytes, but in figure 2F the graphic show a significant decrease in aggrecan RNA levels in ADSC CM compared to ADSCs! Check please. Also, Figure 2K is shown in the text, but is not explained.
Response 9: We thank the reviewer’s comment. We have moved the ADSCs paragraph before the OEC paragraph (new Figure 1). We have explained Figure 2K (new Figure 1L). (page 7, line 320)
Comment 10: The quality of figures is poor.
Response 10: We thank the reviewer’s comment. We have made substantial efforts to improve the figures.
Comment 11: Figure 1 A Authors may include an arrow or a box to indicate the clumped olfactory nasal mucosal tissue
Response 11: We thank the reviewer’s comment. We have included an arrow to indicate the clumped olfactory nasal mucosal tissue. (new Figure 2A)
Comment 12: Figure 1B If I understand, you counted the cells after trypsinization and before the plating. So, how did you not get a nonlinear segment for the OEC#4 sample? Furthermore, error bars should be included in the curve, for example by counting the cells three time.
Response 12: We thank the reviewer’s comment. We have redrawn the figure and added the error bars. (new Figure 2B)
Comment 13: Figure 1D The microphotographs are too small and of low quality. I suggest you reduce the table (Figure 1C) and enlarge the photomicrographs. Moreover, the scale bar of Safranin-O is the same of the ALP and Oil-red O staining (100µm). Please check it.
Response 13: We thank the reviewer’s comment. We followed the reviewer’s suggestions of reducing the table and enlarging the photos. (new Figure 3B)
Comment 14: Figure 2 Microphotographs in G, I, K, L, M, and N show differentiate ADSCs, undifferentiated cells should also be shown.
Response 14: We thank the reviewer’s comment. We are sorry that we did not perform undifferentiated control for ADSC chondrogenesis. We added a picture of the pellet after ADSC’s chondrogenesis (new Figure 1K)
Comment 15: Figure 3 Do the asterisks refer to the comparison between groups and saline ctrl? The comparison should be reported in the figure legend as “ ***p<0.001 compared to the ctrl saline sample”. The mere indication “ctrl:control” is not indicative, also because you have a saline sample ctrl, an OA sample ctrl and a no sample ctrl alone.
Response 15: We thank the reviewer’s comment. We have revised it accordingly. The statement read as”***p<0.001 compared to Saline_Ctrl, OA_OEC, and OA_ADSC.” (new Figure 4)(page 12, line 380)